# Experience and Resilience of Japanese Public Health Nurses during the COVID-19 Pandemic and Their Impact on Burnout

**DOI:** 10.3390/healthcare11081114

**Published:** 2023-04-13

**Authors:** Akari Miyazaki, Tomoko Sankai, Tomoko Omiya

**Affiliations:** 1Program in Nursing Science, Graduate School of Comprehensive Human Science, University of Tsukuba, 1-1-1 Tennodai, Tsukuba 305-8575, Japan; 2Department of Public Health Nursing, Division on Health Innovation and Nursing, Faculty of Medicine, University of Tsukuba, 1-1-1 Tennodai, Tsukuba 305-8575, Japan

**Keywords:** burnout, COVID-19, individual resilience, organizational resilience, public health nurse

## Abstract

Japanese public health nurses (PHNs) at public health centers (PHCs) have played critical roles in infection prevention and control during the COVID-19 pandemic. This study aimed to examine the actual pandemic-related experiences of PHNs and the relation between their experiences, individual resilience, two components of organizational resilience (system and human resilience), and burnout. An analysis of the responses of 351 PHNs revealed that mid-level PHNs scored higher in experience and lower in organizational resilience compared with those in other positions. More than 80% of respondents experienced inappropriate staff allocation. Multiple regression revealed that burnout was positively associated with the components of the experience of PHNs and negatively with individual and human resilience. In hierarchical multiple regression with depersonalization as the dependent variable, the sign of system resilience reversed from negative to positive when human resilience was added. The results highlight the need to prepare for future health crises including establishing a system with enough personnel, promoting human resilience such as collaboration among staff members, and burnout prevention measures, especially among mid-level PHNs. The study also described alternative approaches to comprehend system resilience—namely, a suppression variable of human resilience, promotion of depersonalization, and multicollinearity—and the need for further research on organizational resilience.

## 1. Introduction

The COVID-19 pandemic has so far recorded eight peaks in Japan, with approximately 25.4 million people infected and 50,000 deaths reported as of 5 December 2022 [1]. As local frontline centers for health crisis management, Japanese public health centers (PHCs) are responsible for preventing the occurrence of health crises and building and coordinating local health systems and resources [2]. The current pandemic occurred in the context of continuous cuts to the number of PHCs and their staff as new health issues have emerged since the late 1900s [3,4,5]. During the COVID-19 pandemic, PHCs, as the frontline local quarantine agencies, have been responsible for a variety of public health activities including consultation services, coordination of distribution of medical resources, active epidemiological investigation, health monitoring of patients, and development of the local health care system [6,7].

Among the professionals who have mainly conducted infection control in PHCs are public health nurses (PHNs). Japanese PHNs are nationally licensed nurses specialized in public health. Public health in Japan has been characterized by a focus on local health promotion by PHNs since before World War II. PHNs working in PHCs are involved in infectious diseases, mental health, support for patients with intractable diseases, and disaster countermeasures [8]. PHCs are established by prefectures, designated cities, and special wards in the Tokyo metropolis, and they are distinguished from health centers established by cities. During the pandemic, PHNs at PHCs worked toward pandemic-level infection prevention and control in parallel with these tasks.

Studies have been conducted on health care workers during pandemics including SARS and the current COVID-19 pandemic [9,10,11,12,13]. Meanwhile, only a few studies have targeted public health workers during health crises including pandemic and natural disasters such as floods and earthquakes. In the early stages of the current pandemic, Japanese PHNs faced difficulties in dealing with community residents with negative emotions such as anxiety and anger. It was also distressing to triage suspected patients with limiting testing capacity [14,15]. However, these studies focused on telephone counseling services excluding other services as diverse as those for positive cases, close contacts, and cluster measures. As the pandemic-related attitudes and situations change for health care workers and the relationship between PHCs and medical institutions is constructed in phases, the distress and mental anguish faced by PHNs are changing accordingly [16]. Further research is needed on the distress and mental impact of overall quarantine service on PHNs in Japan.

Burnout is defined as a syndrome of emotional exhaustion in which one’s emotional energy is consumed through providing interpersonal services, resulting in negative and cynical attitudes and feelings toward clients and a negative evaluation toward one’s own work performance [17]. There have been some reports of burnout in health care workers and public servants involved in medical treatment or reconstruction during and post health crises [18,19,20]. In the current pandemic, burnout was observed in approximately 60% of professionals in the public health departments of U.S. government organizations or universities, which was higher than that of health care workers involved in the care of COVID-19-positive individuals [21]. In Japan, 27% of PHC staff including PHNs reported experiencing burnout [22]. Given that burnout is associated with the intention to leave, insomnia, depression, and even diseases such as coronary heart disease and high blood pressure [23,24,25,26], preventing the burnout of PHNs is an urgent issue.

According to a study targeting PHNs involved in providing support to people during the recovery stage of the 2011 Fukushima nuclear accident, PHNs played their role with a sense of mission, which is also possible in the current pandemic [27]. However, there were some movements aimed at changing the working environment, which may reflect that the situation was beyond the capability of their strong sense of mission. All-Japan Prefectural and Municipal Worker’s Unions exist in each prefecture and municipality with about 740,000 workers, accounting for approximately 25% of all prefectural and municipal workers. Japanese Nursing Associations are united with 770,000 nurses, about 60% of all working nurses. These unions and associations have been demanding better work environments at PHCs including securing enough personnel and equipment and higher salaries based on the requests from PHNs to the national and municipal governments [28,29]. Addressing these demands may be effective in radically reforming the public health system and obtaining support for PHNs. Additionally, in 2021, a PHN filed a report against the Labor Standards Inspection Office that accused the municipality of requiring long working hours [30]. Behind all of these remarkable movements is the strong desire of PHNs to change the situation to fulfill their own roles to protect the community. It is critical to determine the predictors of burnout among PHNs for burnout prevention during a pandemic.

Resilience in the face of adversity and disaster has recently received attention as a factor that may mitigate burnout [31,32]. This is defined as the process of bouncing back in the face of adversity, trauma, tragedy, threat, or family or relationship problems, serious health issues, or economic stress [33]. Furthermore, organizational resilience has been gaining attention in recent years, particularly in the field of business. It is defined as an organization’s ability to plan for future crises and adapt to unpredictable emergencies and chronic stresses to mitigate negative outcomes and improve performance in crisis situations [34,35]. In health crises, as PHNs bear a burden with liability to protect people’s health in the frontline in the community, they may be at risk of burnout. PHNs, being under such pressure, which they cannot carry on by their own, point to the need to focus on not only individual resilience but also on organizational resilience to continue fulfilling their roles in crises. Examining the sustainability and effective practices of professional public health organizations is an urgent issue worldwide as society faces various threats such as disasters, terrorism, and infectious diseases.

Our study aimed to examine the actual pandemic-related experiences of PHNs and the association of their experiences, individual resilience, and the two components of organizational resilience of the system and human resilience to burnout. The results are expected to provide important insights into burnout prevention. We also anticipate that these results will ensure that PHCs fully fulfill their functions and roles in the community amid any health crisis in the future.

## 2. Materials and Methods

### 2.1. Participants

In this cross-sectional study, we conducted an anonymous, self-administered questionnaire survey among Japanese PHNs involved in infection prevention and control during the COVID-19 pandemic after January 2020. We distributed both paper- and web-based (Google Forms) questionnaires to 1090 PHNs at 71 PHCs in 29 municipalities. The data were collected from July to December 2022.

### 2.2. Ethical Considerations

This study was approved by the medical ethics review board of the University of Tsukuba (approval no. 1761, 1 July 2022). Participants were informed of the purpose and methods of the study. The participants provided their consent by checking the checkbox on the forms. Personal data including the name of their affiliated PHC and municipality were not collected to secure the anonymity of the participants.

### 2.3. Instruments

#### 2.3.1. Demographic Data

We asked the participants to provide the following information: sex, age, occupational position, career as PHN, family status, and maximum overtime hours per month (Appendix A). The occupational positions were classified as entry-level, mid-level, and administrative positions. In general, entry-level positions include PHNs with less than five years of experience and administrative positions include those with more than 21 years in career or in administrative posts. Mid-level PHNs carry both roles as trainers of entry-level PHNs and assistants to administrative personnel [36].

#### 2.3.2. PHNs’ Experience during the COVID-19 Pandemic

The experiences of PHNs during the COVID-19 pandemic have been unprecedented in terms of novelty and scope—they serve as both nurses and municipal civil servants. We could not identify an appropriate scale to measure their experience of distress and exhaustion with respect to the context. Thus, we extracted items from a qualitative study on distress and its related experiences during the pandemic, which we conducted in 2021 (Appendix A) [37]. This method has been utilized by several studies to obtain practical suggestions that reflect the backgrounds of the sample [38,39,40,41]. The experience of PHNs was measured with 10 items under the question “Did you experience the following situation during the COVID-19 pandemic?” with a five-point Likert scale ranging from 1 = strongly disagree to 5 = strongly agree. Factor analysis revealed three components: “I. Experience of complaints and verbal abuse from patients and community residents” (two items), “II. Experience of weak infrastructural systems for conducting health activities” (four items), and “III. Hardship caused by widespread and prolonged spread of infection” (four items). The reliability values were as follows: Cronbach’s α = 0.797 for the total score, and α = 0.862, 0.651, and 0.630 for components I, II, and III, respectively.

#### 2.3.3. Individual Resilience

We measured individual resilience using the Bidimensional Resilience Scale (BRS) [42] with 21 items. BRS consists of two subscales: innate resilience factor and acquired resilience factor. Each item is rated on a five-point Likert scale ranging from 1 = strongly disagree to 5 = strongly agree; the total score ranges from 21 to 105. The reliability and validity were verified [33]. Cronbach’s α was 0.885 for the total score in our study.

#### 2.3.4. Organizational Resilience

In the field of business, scholars have studied the classification of components of organizational resilience [43] and produced scales such as the Benchmark Resilience Tool [44,45]. However, considering the characteristics of PHCs, which are municipal organizations that are expected to make fair judgment for the public, these scales, which are intended to be used by for-profit organizations, are inappropriate to measure the organizational resilience of PHCs. We opted to extract 11 items from categories on the alleviation of distress for PHNs during the current pandemic in a qualitative study that we conducted in 2021 (Appendix A). By factor analysis, we extracted two components: system resilience (five items) and human resilience (six items). We defined system resilience as “organizational system improvement and system development” and human resilience as “cooperation among staff in the organization and human resources”. Items on system resilience included “There is a situation-specific response by the central government (e.g., establishing new departments, outsourcing of duties, dispatching staff)” and “A shift in working styles in anticipation of long-term pandemics (e.g., introducing an overtime rotation system)”; items on human resilience included “Managers have leadership skills (e.g., quick decision-making, clear instructions, leading the response to difficult cases)” and “Feeling free to share conflicts and personal feelings within the organization”. Items are rated on a five-point Likert scale ranging from 1 = strongly disagree to 5 = strongly agree based on agreement on a list of situations being applicable to one’s organization. The Cronbach’s α = 0.880 for the total score and α = 0.832 and 0.818 for the system and human resilience, respectively.

#### 2.3.5. Burnout

We measured burnout using the Japanese Burnout Scale (JBS) [46]. The JBS consists of three subscales: emotional exhaustion, depersonalization, and reduced personal accomplishment. Emotional exhaustion refers to feelings of being emotionally exhausted from one’s work; depersonalization indicates the impersonal response toward patients and community residents; and personal accomplishment refers to feelings of competence in one’s work [47]. A number of scholars have theorized that personal accomplishment ought to be separated from the other two components of burnout [48]. Indeed, it is often found to have weak correlations with outcomes [49]. Thus, we excluded it from our study. Therefore, we measured “emotional exhaustion” (five items) and “depersonalization” (six items) with the question “How much have you experienced the following situations in the last 6 months?” Respondents indicated their response on a five-point Likert scale ranging from 1 = never to 5 = always. The total score of each subscale was divided by the number of items. The reliability and validity of the scale were verified [46]. In this study, the reliability was verified; the Cronbach’s α = 0.851 and 0.852 for emotional exhaustion and depersonalization, respectively.

### 2.4. Data Analysis

Categorical variables were summarized with percentages (%) and continuous variables as the average and standard deviation. The rates of each item of the experience of PHNs during the pandemic and organizational resilience were summarized with percentages (%). To clarify which demographic data related to the PHNs’ experience and organizational resilience, we compared the mean scores by occupational position using the Kruskal–Wallis test and the Bonferroni’s adjustment method. To detect the association of components of experience, individual resilience, and the two components of organizational resilience to burnout, we conducted single linear and hierarchical multiple regression analyses. We conducted multiple regression analysis hypothesizing that while PHNs were negatively affected by the pandemic, the presence of individual resilience and components of organizational resilience mitigates their burnout, and allows them to continue fulfilling their roles without leaving their jobs. The residual variance of experience II and III showed a normal distribution. However, “I. Experience of complaints and verbal abuse from patients and community residents” was not entered in the multiple regression analysis due to the ceiling effect. The components of the experience of the PHNs were added separately in different patterns to avoid multicollinearity.

All analyses were conducted using IBM SPSS Statistics for Windows, Version 28.0 (IBM Japan, Tokyo, Japan), and a *p* value of < 0.05 was considered significant.

## 3. Results

### 3.1. General Demographics of the Participants

A total of 365 PHNs (33.5%) responded to the survey and 351 (96.2%) completed more than half of the items and provided their consent. The participants’ demographic characteristics are shown in Table 1.

The mean scores for individual resilience and two components of burnout—emotional exhaustion and depersonalization—are shown in Table 2.

### 3.2. PHNs’ Experience and Organizational Resilience Perceived by PHNs

Table 3 shows the rating for each PHN experience item. We evaluated PHNs who selected strongly agree and agree as those who perceived the items as applicable to their experiences. More than 85% of PHNs had “I. Experience of complaints and verbal abuse from patients and community residents”. Regarding “II. Experience of weak infrastructural systems for conducting health activities”, more than 80% experienced “Inappropriate staff allocation from outside and within PHCs”, and about 90% experienced “Overtime/holiday work changes the life pattern”. Related to “III. Hardship caused by widespread and prolonged spread of infection”, 82% faced “Difficulty in coordinating medical care for patients owing to lack of medical resources”, and 94% perceived “Without the concept of “full beds”, duties of dealing with patients are endless”.

Table 4 shows the evaluated score rate for each item of organizational resilience. We evaluated PHNs who selected strongly agree and agree as those who perceived the items as applicable to their organization, and PHNs who selected disagree and strongly disagree as those who perceived the items as non-applicable. Among the system resilience items, “There is a situation-specific response by the central government” was the item that most PHNs perceived to be applicable to their own organizations. Meanwhile, a similar percentage (about 37%) of PHNs evaluated the item “A shift in working styles in anticipation of long-term pandemics” as applicable and non-applicable to their PHCs. As for human resilience, more than 65% of PHNs agreed that “Managers have leadership skills” and “There is a PHN who assists and supports the PHN manager”. 

A comparison of the mean scores of experiences and organizational resilience by occupational position is shown in Table 5. Mid-level PHNs tended to evaluate the experience items higher, and organizational resilience items lower.

### 3.3. Association of PHNs’ Experience, Individual Resilience, and Organizational Resilience with Emotional Exhaustion

Table 6 gives the results of our single linear and hierarchical multiple regression analyses with emotional exhaustion as the dependent variable. By the single linear regression analysis, experiences II, “Experience of weak infrastructural systems for conducting health activities” (β = 0.335, *p* < 0.001) and III “Hardship caused by widespread and prolonged spread of infection” (β = 0.215, *p* < 0.001) showed a positive association with emotional exhaustion. Individual (β = −0.278, *p* < 0.001), system (β = −0.157, *p* < 0.05), and human resilience (β = −0.276, *p* < 0.001) were negatively associated with emotional exhaustion.

The multiple regression analysis showed that the PHNs’ experience of II (β = 0.279, *p* < 0.001) and III (β = 0.219, *p* < 0.001) were positively associated with emotional exhaustion. Individual (β = −0.199, *p* < 0.001; β = −0.213, *p* < 0.001) and human resilience (β = −0.220, *p* = 0.001; β = −0.271, *p* < 0.001) showed a negative association. Human resilience also showed a larger effect than individual resilience. The control variable, holding an administrative position in reference to a mid-level position, was negatively associated, with experience III as an independent variable (β = −0.127, *p* < 0.05).

### 3.4. Association of PHNs’ Experience, Individual Resilience, and Organizational Resilience with Depersonalization

Table 7 gives the results of the single linear and hierarchical multiple regression analyses with depersonalization as the dependent variable. Through the single linear regression analysis, the PHNs’ experiences of II (β = 0.356, *p* < 0.001) and III (β = 0.184, *p* < 0.001) showed a positive association with depersonalization, whereas individual (β = −0.295, *p* < 0.001), system (β = −0.136, *p* < 0.05), and human resilience (β= −0.302, *p* < 0.001) showed a significant negative association.

In both patterns with experiences II and III as independent variables, the multiple regression analysis showed that experiences II (β = 0.292, *p* < 0.001) and III (β = 0.166, *p* = 0.001) had a positive association with depersonalization. Individual (β = −0.194, *p* < 0.001; β = −0.211, *p* < 0.001) and human resilience (β = −0.298, *p* < 0.001; β = −0.349, *p* < 0.001) showed negative associations. Human resilience showed a larger effect than the components of experience and individual resilience. Although system resilience showed no significant association in Model 1, it showed a positive association after adding human resilience in Model 2 in both patterns, with experiences II and III as the independent variables (β = 0.167, *p* < 0.01; β = 0.142, *p* < 0.05).

## 4. Discussion

This study targeted Japanese PHNs working in disease prevention and control during the COVID-19 pandemic as community frontline personnel. We examined the actual state of their experience and organizational resilience as well as the association of their experience, individual resilience, and organizational resilience to burnout.

### 4.1. Actual State of Individual Resilience and Burnout

The mean score for resilience was 74.56, much higher compared with that reported in other studies using the same tool: 70.0 in the study targeting municipal civil servants involved in the Great East Japan Earthquake and flood disaster in 2011 [50], and 71.1 among nurses working in hospitals [51]. The higher mean value indicates two possibilities: PHNs with higher individual resilience tend to hold onto their roles as PHNs continuously, and their individual resilience may have improved through their experiences as PHNs. Regarding the former, considering that their working environment is not adaptive to those not individually highly resilient, the organization must improve its ability to adapt to health crises. Individual resilience is both innate and acquired [42], and is associated with positive emotions, high self-efficacy, a strong social support system, self-knowledge, and satisfaction [52,53,54]. To maintain the individual resilience of PHNs’ under a health crisis, PHCs should clarify the roles of PHNs and reallocate PHNs in appropriate positions to utilize their specialty and cooperation with other departments and the community. Additionally, introducing intervention programs such as stress management and resiliency training and the Penn Resiliency Program may help maintain and improve their individual resilience [55,56].

The average scores of the subscales of burnout were 3.36 for emotional exhaustion and 2.36 for depersonalization. The PHNs in our study scored higher than the following samples in other studies conducted using the same tool in Japan: neurologists [57], community support center staff including PHNs [58], and psychiatric nurses [59]. The higher burnout score in our sample suggests that their particular experience during the pandemic may have promoted emotional resource consumption and led them to provide an impersonal response to community residents and medical workers.

### 4.2. Actual State of the PHNs’ Experience during the Pandemic

Our findings revealed the actual state of experience and organizational resilience perceived by PHNs during the current pandemic. More than 85% of PHNs experienced complaints and verbal abuse from patients and community residents, and it was experienced through all phases of the pandemic. The qualitative study by Usukura et al. [15] showed that at the beginning of the pandemic, PHCs were treated as “a complaints counter instead of a health consultations provider”. During the COVID-19 pandemic, people experienced negative emotions such as anger, fear, and anxiety [60,61,62], which were expressed strongly as verbal abuse and complaints toward PHNs. This scenario will definitely be repeated in future health crises. Meanwhile, not all PHNs have the individual capability to accept strong emotions from community residents. Thus, there is an urgent need to establish an organizational system that can emotionally assist them through sharing their common issues and emotions and offering personal support. Moreover, as a fundamental resolution, issuing an ordinance to provide a warning and legal move to residents against unreasonable verbal abuse and its dissemination to the wider public may be effective in protecting workers; however, only a few municipalities have such an ordinance in effect [63]. The announcement of “this telephone message is recorded” before the conversation to restrain callers and prevent fraud, which are officially recommended by police departments and used widely, can be implemented at PHCs to prevent unreasonable verbal abuse [64].

Experience II included the irritation caused by the circumstance of having insufficient organizational basis for conducting health activities in spite of the disaster-level situation. The survey targeting administrative personnel in PHCs showed that about 90% of PHCs changed staff allocations within the PHCs and municipal organizations during the pandemic [65]. However, our survey revealed that more than 80% of PHNs deemed the staff allocation insufficient. Thus, even if organizations develop measures to deal with the situation, the degree of effort can be insufficient for PHNs who are working on the frontline. This shows that the managers who decide policies for PHCs must listen to the PHNs regularly working on the frontline to implement adequate measures during each phase of the pandemic.

Experience III indicates a situation that is impossible to change by the PHNs themselves. Almost all of the PHNs experienced the uncertainty of the pandemic, which may have elicited anxiety [66]. Indeed, more than 80% of PHNs across Japan struggled with medical coordination with insufficient medical resources. The Ministry of Health, Labor and Welfare reported cases where more than 36,000 positive cases were awaiting medical coordination or confirmation by PHCs at the peak of the fifth wave (July to September 2021) and where more than 80% of beds for COVID-19 were occupied at the peak of the sixth wave (January to March 2022) [67,68]. They struggled in fulfilling their responsibility in the proper distribution of medical resources through being an intermediary between medical institutions and patients. They also struggled when the demand far exceeded the supply in medical resources.

Regarding occupational positions, mid-level PHNs tended to evaluate their experience higher compared with those in administrative positions, and organizational resilience was lower compared with those in entry-level and administrative positions. The results of the multiple regression analysis showed that mid-level PHNs tended to have reached burnout compared to the administrative personnel. As mid-level PHNs have enough experience to conduct health activities, they help train entry-level PHNs, assist managers, and serve as practical leaders in PHCs [36,69]. Therefore, mid-level PHNs may especially be expected to act as immediate assets, and the heavy workload and responsibility stressed them. Additionally, most mid-level PHNs are in the life stages of maintaining marriages, giving birth, and raising children. In May 2020, after the declaration of a state of emergency by the Japanese government, 86% of schools canceled all classes and 59% of companies adopted teleworking [70,71], which may have changed their family’s lifestyle. Research has shown that work–family conflict is positively related to burnout [72,73]. However, the particularity of mid-level PHNs have not yet been revealed, warranting further investigation. The mid-level PHNs in our study may have perceived their experience to be more severe. As they may always be conscious of conducting duties in efficient ways, they may also have evaluated organizational systems as being more vulnerable. In future crises, PHNs will face panicked community residents amid new threats and a seemingly endless situation. Their burden has to be reduced through system development, collaboration with the community, and staffing personnel distributions. In particular, it is crucial to lessen the burden to prevent burnout in mid-level PHNs and stabilize the organization by actively and broadly incorporating their viewpoints and considerations. Specifically, participation in unions or associations that can raise the issue with national and municipal governments is one way for an individual worker to protect themselves. Furthermore, as an organization, a PHC should provide opportunities for PHNs to share their problems and distress in a psychologically safe environment [74]. This will enable managers, who decide the policies at PHCs, to notice other ideas and incorporate them into policy, which may result in a more sustainable organization.

### 4.3. Relation of PHNs’ Experience, Resilience, and Organizational Resilience to Burnout

The result of the single linear regression analysis showed that both components of the experience of PHNs had a positive association with burnout. We found negative associations with individual and organizational resilience. Thus, the components of the PHNs’ experience during the pandemic may have possibly promoted burnout, although individual and organizational resilience may have functioned as moderators. To examine the relations within multiple variables, we conducted hierarchical multiple regression analysis.

#### 4.3.1. Relation of PHNs’ Experience, Resilience, Organizational Resilience to Emotional Exhaustion

In the pattern with emotional exhaustion as the dependent variable, both experiences II and III were positively associated, and individual and human resilience were negatively associated with emotional exhaustion. No significant association was found for system resilience. Thus, experiences II and III may promote emotional exhaustion, whereas individual and human resilience may inhibit it.

Kubo [75] noted that emotional resources were more likely to be depleted in “dedicated workers who tend to be deeply troubled by what they cannot do”. The PHNs’ experience of II and III were due to the characteristics of the pandemic or a fragile organization that would never be improved and could hardly be controlled by the PHNs themselves. However, trying to devote their efforts and consume their emotional resources to cope with and adapt to these uncontrollable matters could then lead to emotional exhaustion.

Human resilience can also be described as the population’s ability to overcome adversity in an organization. We found that the presence of human resilience may play an important role in preventing emotional exhaustion in crises. Moreover, as human resilience showed a larger effect than individual resilience, it has the potential to prevent emotional exhaustion without relying on individual resilience. Considering the items of human resilience, emotional exhaustion may be alleviated by a manager with clear instructions that provide as much perspective as possible in uncertain situations, the review and reconstruction of human resources to reduce the burden concentrated on a few PHNs, and a commitment by all staff members in the PHC. 

#### 4.3.2. Relation of PHNs’ Experience, Individual Resilience, Organizational Resilience to Depersonalization

In the pattern with depersonalization as the dependent variable, experiences II and III were positively associated with depersonalization. Individual and human resilience showed negative associations. In other words, experiences II and III promoted depersonalization, whereas individual and human resilience may reduce it.

The PHNs’ experience of III is difficult for PHNs to control, but may be repeated in future pandemics. Leiter et al. [76] showed that support from supervisors and colleagues could reduce depersonalization, and Takahashi et al. [59] showed that emotional support among nurses who had experienced similar situations could alleviate depersonalization. Our study showed that even when PHNs struggled to address the uncontrollable circumstances and emotional distress, an environment in which the PHNs could feel free to express their emotions and emotional support such as concern for individual health by managers may reduce depersonalization. Depersonalization may occur not only because of the concentrated burden on some staff members in weak infrastructural systems, but also because of the frustration of the weak organization itself and the sense of irritation toward the organization. Indeed, the presence of a supervisor who adjusts the workload of the entire organization and plays a supervisory role in providing advice and consultation can inhibit depersonalization [58].

As in the case with emotional exhaustion as the dependent variable, we found that human resilience may play an important role in preventing depersonalization. For example, the presence of a PHN who assists managers may provide a bird’s-eye view of the organization from multiple perspectives and emotional support to the managers, which may be beneficial to prevent depersonalization toward coworkers. Furthermore, human resilience showed a greater impact on depersonalization compared with experience and individual resilience. Thus, enhancing human resilience in organizations may be effective to prevent depersonalization without relying on individual resilience alone.

Although system resilience showed a significant negative correlation with depersonalization in the single linear regression analysis, the result of the hierarchical multiple regression analysis showed a significantly positive association when human resilience was added in Model 2. We examined two possible explanations for this phenomenon: system resilience being a suppressor variable or promoter of depersonalization.

First, we found a moderate positive correlation between the two components of organizational resilience (r = 0.63, *p* < 0.001), and a stronger correlation between depersonalization and human resilience (r = −0.31, *p* < 0.001) than that between depersonalization and system resilience (r = −0.14, *p* < 0.05). Based on these correlation results, and the reversed sign, system resilience may have acted as a suppression variable (net suppression) for human resilience. In this case, human resilience may be more effective in reducing depersonalization in situations where system resilience is low. A study on the work environments of nurses showed that nurses with high perceptions toward resources such as good teamwork and recognition from supervisors could mitigate burnout compared with those whose perception of having resources was low, but only when quantitative demands increased [77]. We observed similar results in our study.

Given the tendency for system resilience to have a reversed sign, establishing a system within the organization can be a burden for some individuals. When they improved system resilience by establishing and controlling the organizational system, they may face new challenges and difficulties to deal with, which may promote depersonalization. Alternatively, considering that it was only observed when the dependent variable was depersonalization, the improvement in system resilience and the promotion of depersonalization may have occurred simultaneously, according to the increase in confirmed COVID-19 cases. When confirmed cases rise drastically, depersonalization can be promoted by recognizing COVID-19-positive individuals not as objects of nursing care but as objects of work. Meanwhile, the development of tools, introduction of support staff, and adoption of a rotation system in organizations are measures that are necessary when the number of confirmed cases becomes large. In this case, this tendency may be applied, particularly among PHNs in areas with a large number of confirmed cases. However, as we did not obtain the number of confirmed cases in their areas, we could not verify this particular point.

In this study, we measured organizational resilience with items whose validity and accuracy were not ensured. Moreover, the concept of the organizational resilience of PHCs in a pandemic has not been established. However, considering the multifaceted situation of PHCs in the pandemic, both the possibilities—acting as suppression variable and promoter of depersonalization—cannot be denied. Further exploration of the concept of organizational resilience of PHCs during a pandemic is identified as a future challenge.

### 4.4. Limitations

This study had several limitations. First, the validity and precision of the items to measure the PHNs’ experience and components of organizational resilience were not ensured, and the reliability coefficients and factor loadings were low in the numerical results of the analysis. In hierarchical regression analysis, the possibility of multicollinearity could not be ruled out. Although the variance inflation factor values showed no multicollinearity of system and human resilience, the condition indicator and ratio of variance showed suspected multicollinearity. Further research to examine the concepts and scales of the PHNs’ experience and organizational resilience at PHCs during a pandemic will allow us to obtain clearer indications. Second, as this study is a cross-sectional study, causal relations were unclear. Third, the response rate in this study was lower than that of a pre-pandemic study targeting Japanese PHNs [78,79]. The survey was conducted during the busy period of the pandemic, and PHNs who perceived strong difficulties in dealing with their duties and strong dissatisfaction with the organization and the government may have been more motivated to respond to the survey. This may have led to higher negative experience and lower organizational resilience ratings compared with the population. Finally, recall bias and not having included PHNs who left their jobs due to burnout or who belonged to municipalities with relatively small numbers of confirmed cases were also limitations of the study.

## 5. Conclusions

PHNs at PHCs experience difficult situations because of the complaints and verbal abuse from patients and community residents, weak infrastructural systems, and widespread and prolonged spread of infection. Considering that most PHNs experienced inappropriate staff allocation, it is critical to ensure sufficient personnel in PHCs located in a community undergoing an outbreak during a future crisis. Mid-level PHNs scored worse for experiences and the components of organizational resilience compared with personnel in other positions. Multiple regression analysis revealed that two components of experience, namely, the experience of weak infrastructural systems for conducting health activities and hardship caused by the widespread and prolonged spread of infection, were positively associated with burnout, whereas individual and human resilience showed a negative association. When depersonalization was the dependent variable, system resilience, which showed a negative association in the single linear regression analysis, demonstrated a positive association when human resilience was added in Model 2. System resilience may have become a suppression variable of human resilience, suggesting that human resilience may have a larger contribution to inhibiting depersonalization in situations where system resilience is low. Alternatively, as system resilience is prone to sign reversal, then the establishment of a system in an organization may become a burden for PHNs and thereby promote depersonalization.

In future pandemics, PHNs will face the same experiences of complaints from community residents and the widespread and prolonged spread of infection. Given that mid-level PHNs tend to face more difficulties and perceive the fragility of organizational resilience, their opinions and considerations should be incorporated widely and actively in designing measures. Moreover, to ease the burden of PHNs, personnel arrangement should include securing human resources and sharing roles. It is important to promote preparedness through the organizational system, especially human resilience, in the form of cooperation among staff and great leadership. Further research on the components of organizational resilience of PHCs, especially system resilience, could reveal valuable insights for the preparedness of public health organizations for health crises.

## Figures and Tables

**Table 1 healthcare-11-01114-t001:** Socio-demographic data of the respondents.

Variables (N)	Categories	n (%) or M ± SD	
Sex (N = 351)	Male	32	(9.1)
	Female	318	(90.6)
	Other	1	(0.3)
Age (N = 349)	(M ± SD)	39.21 ± 12.41	
	20 s	107	(29.9)
	30 s	90	(26.6)
	40 s	58	(16.9)
	≥50 s	94	(5.7)
Occupational position (N = 351)	Entry-level	139	(39.6)
	Mid-level	113	(32.2)
	Administrative position	99	(28.2)
Career as PHN (N = 348)	<10 years	180	(51.7)
	10–<20 years	60	(17.2)
	20–<30 years	41	(11.8)
	≥30 years	67	(19.3)
Children below middle school (N = 349)	Yes	72	(20.6)
	No	277	(79.4)
Family member in need of nursing care (N = 350)	Yes	20	(5.7)
	No	330	(94.3)
Maximum overtime hours per month (N = 343)	<50 h	66	(19.2)
	50–<100 h	130	(37.9)
	100–<150 h	99	(28.9)
	≥150 h	48	(14.0)

**Table 2 healthcare-11-01114-t002:** Scores for individual resilience and burnout.

Variables	N	M ± SD	Range	
			Potential	Actual
Individual resilience (total)	342	74.56 ± 10.22	21–105	37–101
Burnout				
Emotional exhaustion	350	3.36 ± 0.99	1.00–5.00	1.00–5.00
Depersonalization	344	2.36 ± 0.89	1.00–5.00	1.00–5.00

**Table 3 healthcare-11-01114-t003:** Experience rating of each item of the PHN experience during the COVID-19 pandemic.

Items	N	Strongly Agree		Agree		Neutral		Disagree		Strongly Disagree	
		n	(%)	n	(%)	n	(%)	n	(%)	n	(%)
Experience I: Experience of complaints and verbal abuse from patients and community residents											
Complaints and verbal abuse from community residents about the PHC’s response	351	215	(61.3)	110	(31.3)	15	(4.3)	9	(2.6)	2	(0.6)
Unreasonable demands or long hours of complaints from COVID-19-positive patients	351	211	(60.1)	99	(28.2)	27	(7.7)	12	(3.4)	2	(0.6)
Experience II: Experience of weak infrastructural systems for conducting health activities											
Difficulty in gaining understanding of requests for cooperation in preventing the spread of infection from close contacts without a legal basis	349	95	(27.2)	136	(39.0)	75	(21.5)	37	(10.6)	6	(1.7)
Inappropriate staff allocation from outside and within PHCs	350	182	(52.0)	105	(30.0)	41	(11.7)	20	(5.7)	2	(0.6)
Lack of uniformity within the PHC (e.g., decisions on dealing with community residents, awareness of tasks, overall flow)	349	106	(30.4)	124	(35.5)	67	(19.2)	44	(12.6)	8	(2.3)
Overtime/holiday work changes the life pattern	350	259	(74.0)	68	(19.4)	15	(4.3)	6	(1.7)	2	(0.6)
Experience III: Hardship caused by widespread and prolonged spread of infection											
Difficulty in coordinating medical care for patients owing to lack of medical resources	351	184	(52.4)	107	(30.5)	28	(8.0)	20	(5.7)	12	(3.4)
Without the concept of “full beds”, duties of dealing with patients are endless	349	245	(70.2)	83	(23.8)	13	(3.7)	5	(1.4)	3	(0.9)
Difficulty in providing smooth health instructions to foreign community residents	350	87	(24.9)	135	(38.6)	66	(18.9)	46	(13.1)	16	(4.6)
Being occupied with medical management for individuals, incapable of performing important duties for preventing infection spread	351	154	(43.9)	121	(34.5)	47	(13.4)	23	(6.6)	6	(1.7)

**Table 4 healthcare-11-01114-t004:** Perceived organizational resilience rate of each item.

Items	N	Strongly Agree		Agree		Neutral		Disagree		Strongly Disagree	
		n	(%)	n	(%)	n	(%)	n	(%)	n	(%)
System Resilience											
There is a situation-specific response by the central government (e.g., establishing new departments, outsourcing of duties, dispatching staff)	349	75	(21.5)	163	(46.7)	58	(16.6)	40	(11.5)	13	(3.7)
Support staff from outside of PHCs is kept fixed for the long-term and allows an accurate response to tasks	348	26	(7.5)	124	(35.6)	89	(25.6)	87	(25.0)	22	(6.3)
In-office systems are constantly made and improved as necessary (e.g., developing tools for sharing information)	348	38	(10.9)	151	(43.4)	79	(22.7)	57	(16.4)	23	(6.6)
A system for receiving support staff into PHCs is well prepared and coordinated as appropriate	348	35	(10.1)	163	(46.8)	77	(22.1)	61	(17.5)	12	(3.4)
A shift in working styles in anticipation of long-term pandemics (e.g., introducing an overtime rotation system)	348	40	(11.5)	88	(25.3)	93	(26.7)	73	(21.0)	54	(15.5)
Human Resilience											
Managers have leadership skills (e.g., quick decision-making, clear instructions, leading the response to difficult cases)	349	93	(26.6)	148	(42.4)	65	(18.6)	34	(9.7)	9	(2.6)
There is a PHN who assists and supports the PHN manager	349	76	(21.8)	151	(43.3)	69	(19.8)	43	(12.3)	10	(2.9)
Managers create a comfortable working environment. (e.g., culture of openness, careful attention to individual health)	347	37	(10.7)	127	(36.6)	101	(29.1)	50	(14.4)	32	(9.2)
In responding to emergencies, the arrangement and management of work within the PHC organization are reviewed	349	58	(16.6)	149	(42.7)	62	(17.8)	57	(16.3)	23	(6.6)
Feeling free to share conflicts and personal feelings within the organization	348	53	(15.2)	125	(35.9)	85	(24.4)	59	(17.0)	26	(7.5)
There is a sense of awareness and common understanding of the disaster response to be undertaken by all staff at PHCs	348	31	(8.9)	115	(33.0)	104	(29.9)	67	(19.3)	31	(8.9)

**Table 5 healthcare-11-01114-t005:** Differences in the mean score for experience and organizational resilience by occupational position.

	Entry-Level		Mid-Level		Administrative Position		*H* ^1^ (*p*)
	*n*	M ± SD	*n*	M ± SD	*n*	M ± SD	
Experience (total)	136	41.68 ± 5.44	112	43.23 ± 4.94	95	41.35 ± 5.87	7.61 (0.022) ^A^
I. Experience of complaints and verbal abuse from patients and community residents	139	8.97 ± 1.39	113	9.12 ± 1.43	99	8.70 ± 1.61	5.03 (0.081)
II. Experience of weak infrastructural systems for conducting health activities	137	16.17 ± 2.87	113	17.11 ± 2.16	95	16.25 ± 2.63	6.87 (0.032)
III. Hardship caused by widespread and prolonged spread of infection	138	16.55 ± 2.47	112	16.99 ± 2.61	98	16.38 ± 2.98	3.94 (0.139)
Organizational resilience (total)	136	38.65 ± 7.00	113	35.06 ± 8.64	96	37.83 ± 8.56	11.93 (0.003) ^B,C^
System resilience	136	17.04 ± 3.88	113	15.57 ± 4.57	98	17.08 ± 4.11	8.43 (0.015) ^B^
Human resilience	136	21.61 ± 4.15	113	19.50 ± 4.90	97	20.76 ± 5.15	13.22 (0.001) ^B^

^1^ *H* = Kruskal–Wallis test. ^A^ Mid-level evaluated higher than administrative position. ^B^ Mid-level position lower than entry-level. ^C^ Mid-level evaluated lower than administrative position. Pairwise comparisons between occupational positions by Bonferroni method; only differences with *p* < 0.05 are marked with capital letters.

**Table 6 healthcare-11-01114-t006:** Hierarchical multiple regression analysis with emotional exhaustion as the dependent variable, and experience, individual resilience, and organizational resilience as the independent variables (N = 351).

Variables ^1^	Regression Coefficient		Model 1		Model 2		Model 1		Model 2	
	*β*	*p*	*β*	*p*	*β*	*p*	*β*	*p*	*Β*	*p*
Control										
Sex (Female = 1)	−0.010	0.854	0.025	0.611	0.014	0.768	0.050	0.324	0.037	0.458
Children below middle school	−0.087	0.104	−0.149	0.006	−0.147	0.006	−0.161	0.004	−0.160	0.003
Family member in need of nursing care	−0.084	0.115	−0.078	0.115	−0.072	0.144	−0.081	0.110	−0.075	0.134
Maximum overtime hours	0.016	0.764	−0.001	0.979	−0.006	0.906	0.011	0.839	0.002	0.966
Occupational Position (mid-level as reference)										
Entry-level	0.038	0.478	−0.039	0.526	−0.016	0.788	−0.068	0.274	−0.037	0.540
Administrative position	−0.135	0.011	−0.111	0.080	−0.110	0.078	−0.131	0.043	−0.127	0.045
PHNs’ experience										
II. Experience of weak infrastructural systems for conducting health activities	0.335	<0.001	0.300	<0.001	0.279	<0.001	-	-	-	-
III. Hardship caused by widespread and prolonged spread of infection	0.215	<0.001	-	-	-	-	0.212	<0.001	0.219	<0.001
Individual resilience	−0.278	<0.001	−0.220	<0.001	−0.199	<0.001	−0.242	<0.001	−0.213	<0.001
Organizational resilience										
System resilience	−0.157	0.003	−0.041	0.420	0.086	0.166	−0.096	0.061	0.068	0.277
Human resilience	−0.276	<0.001	-	-	−0.220	0.001	-	-	−0.271	<0.001
*R^2^*			0.201	<0.001	0.229	<0.001	0.163	<0.001	0.206	<0.001
Δ*R^2^*					0.028	<0.001			0.043	<0.001

^1^ In Model 1, control variables, components of the PHNs’ experience, individual resilience, and system resilience were added. Human resilience was added in Model 2. Forced entry method and mean substitution for missing data were conducted.

**Table 7 healthcare-11-01114-t007:** Hierarchical multiple regression analysis with depersonalization as the dependent variable, and experience, individual resilience, and organizational resilience as the independent variables (N = 351).

Variables	Regression Coefficient		Model 1		Model 2		Model 1		Model 2	
	*β*	*p*	*β*	*p*	*β*	*p*	*β*	*p*	*β*	*p*
Control										
Sex (Female = 1)	−0.129	0.016	−0.087	0.074	−0.102	0.032	−0.066	0.194	−0.084	0.090
Children below middle school	−0.035	0.516	−0.103	0.051	−0.101	0.050	−0.108	0.053	−0.107	0.044
Family member in need of nursing care	−0.019	0.725	0.009	0.852	0.018	0.699	0.011	0.836	0.019	0.692
Maximum overtime hours	0.018	0.741	0.017	0.735	0.011	0.824	0.035	0.509	0.024	0.637
Occupational Position (mid-level as reference)										
Entry-level	0.060	0.266	−0.038	0.526	−0.008	0.897	−0.071	0.258	−0.031	0.605
Administrative position	−0.205	<0.001	−0.171	0.006	−0.170	0.005	−0.196	0.003	−0.190	0.002
PHNs’ experience										
II Experience of weak infrastructural systems for conducting health activities	0.356	<0.001	0.321	<0.001	0.292	<0.001	-	-	-	-
III Hardship caused by widespread and prolonged spread of infection	0.184	<0.001	-	-	-	-	0.156	0.002	0.166	0.001
Individual resilience	−0.295	<0.001	−0.223	<0.001	−0.194	<0.001	−0.248	<0.001	−0.211	<0.001
Organizational resilience										
System resilience	−0.136	0.011	−0.006	0.910	0.167	0.006	−0.069	0.175	0.142	0.023
Human resilience	−0.302	<0.001	-	-	−0.298	<0.001	-	-	−0.349	<0.001
*R^2^*			0.229	<0.001	0.279	<0.001	0.160	<0.001	0.230	<0.001
Δ*R^2^*					0.051	<0.001			0.070	<0.001

Note: In Model 1, control variables, components of the PHNs’ experience, individual resilience, and system resilience were added. Human resilience was added in Model 2. Forced entry method and mean substitution for missing data were conducted.

## Data Availability

As this study targeted public officials who are deeply involved in public administration, the data presented in this research are not available.

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
