# Peer review of "Experience and Resilience of Japanese Public Health Nurses during the COVID-19 Pandemic and Their Impact on Burnout"

_healthcare, 2023, doi:10.3390/healthcare11081114_

Round 1

Reviewer 1 Report

-Introduction

1. The harmfulness of job burnout can be further discussed in Line 73-73, which has led to the study of the causes of job burnout. In fact, a number of studies have proven that job burnout is related to psychological factors. A new starting point is needed.

2. Is it appropriate to include the research hypothesis in Lines 92 to 95? It can be interspersed with the above, but only the purpose and significance of the research are discussed here.

-Instruments

1. In Part 2.2.4, I observed that you conducted a qualitative study and extracted part of it for measurement. Maybe you can list a few more items

2. The ethical statement can be moved below 2.1

3. When the author explored this problem, there was no qualitative, operable and standard measurement method for some variables, which I also considered and worried about, although the author had mentioned it in the limitation.

-Tables

1. Many tables do not conform to academic norms, and the position of Table7 in the article is not quite right

2. Is "other" in the gender column of Table 1 a vacant value?

-Discussion

In Part 1. 4.1, lines 290-292, the resilience of individuals can be improved not only by the intervention of the external factor of relocation or redistribution, but also by other intervention strategies.

Author Response

15 March 2023

Dear Dr. Yolande Cai

And Reviewers

We sincerely express our gratitude to the Editor and the reviewers who took valuable time and reviewed the manuscript with significant comments. We describe below our point-by-point response to all the comments of the reviewers.

Sincerely,

Akari Miyazaki

Point-by-point response to the reviewers’ comments

Reviewer1

-Introduction

1. The harmfulness of job burnout can be further discussed in Line 73-73, which has led to the study of the causes of job burnout. In fact, a number of studies have proven that job burnout is related to psychological factors. A new starting point is needed.

Response:

Thank you for your valuable comment.

We added some description about the effect of burnout as follows;

“Given that burnout is associated with the intention to leave, insomnia, depression, and even diseases such as coronary heart disease and high blood pressure [23-26], preventing PHNs’ burnout is an urgent issue.”

2. Is it appropriate to include the research hypothesis in Lines 92 to 95? It can be interspersed with the above, but only the purpose and significance of the research are discussed here.

Response:

We appreciate your thoughtful comment.

We moved hypothesis to 2.4. Data Analysis in the explanation of multiple regression analysis.

“We conducted multiple regression analysis hypothesizing that while PHNs were negatively affected by the pandemic, the presence of individual resilience and components of organizational resilience mitigate their burnout and allow them to continue fulfilling their roles without leaving their jobs.”

-Instruments

1. In Part 2.2.4, I observed that you conducted a qualitative study and extracted part of it for measurement. Maybe you can list a few more items

Response:

Thank you very much for your valuable comment. As you suggested, we showed some of the items in 2.3.4 Organizational Resilience as follows. Also, as other reviewer suggested, we added a copy of questionnaire as an appendix to show the detail.

“Items on system resilience included “There is a situation-specific response by the central government (e.g., establishing new departments, outsourcing of duties, dispatching staff)” and “A shift in working styles in anticipation of long-term pandemics (e.g., introducing an overtime rotation system)”; items on human resilience included “Managers have leadership skills (e.g., quick decision-making, clear instructions, leading the response to difficult cases)” and “Feeling free to share conflicts and personal feelings within the organization.””

2. The ethical statement can be moved below 2.1

Response:

Thank you for your comment. As you refer, we moved ethical consideration below 2.1.

3. When the author explored this problem, there was no qualitative, operable and standard measurement method for some variables, which I also considered and worried about, although the author had mentioned it in the limitation.

Response:

We appreciate your meaningful comment.

We agree the concern what you have pointed out. As you refer, this was the greatest concern also for us. Thus, we repeated discussions and we concluded that one of the most important and valuable point of our study is to investigate in the midst of the pandemic, where PHNs were still struggling. It is critical because this event have to be recorded and investigated before fading from their memories for future pandemic. Learning from the history may be critical and our study may be able to contribute to the preparation for future crisis in all over the world. It is our faith to contribute to the future through our study, considering that the experience of past pandemics was not well utilized and reflected to our present public health system in Japan. However, as your consideration, further investigation is definitely required.

-Tables

1. Many tables do not conform to academic norms, and the position of Table7 in the article is not quite right

Response:

Thank you for your comment.

We made some modification on tables and its position.

2. Is "other" in the gender column of Table 1 a vacant value?

Response:

No, there was 1 (0.3%) who answered “other” in the gender, and we showed it in table.

-Discussion

In Part 1. 4.1, lines 290-292, the resilience of individuals can be improved not only by the intervention of the external factor of relocation or redistribution, but also by other intervention strategies.

Response:

Thank you for your valuable suggestion. We had made discussion only about the external factor of working environment. However, as you refer, intervention to maintain and improve the individual resilience is studied, so we added some description of individual intervention as follows;

“Additionally, introducing intervention programs such as Stress Management and Resiliency Training and the Penn Resiliency Program may help maintain and improve their individual resilience [53,54].”

Reviewer 2 Report

Dear authors, 

Your study is timely and addresses an issue of great practical importance for occupational safety and health, as well as public health more broadly. That is, preventing burnout among Public Health Nurses (PHNs), who are essential to addressing worldwide threats from "disasters, terrorism, and infectious disease." As you note in the Introduction, I wholeheartedly agree that "PHNs being under such pressure, which they cannot carry on their on their own, points to the need to focus on not only individual resilience but also organizational resilience to continue fulfilling their roles in crises."

Your well-written manuscript is excellent, with an appropriate research design, methods that are clearly described, results that are clearly presented, and conclusions that are supported by the results. Prior to publication, I would like to offer a few minor questions, comments, and suggestions for your consideration, in order to further strengthen the manuscript. Below, these comments are organized by manuscript section. 

Introduction

You provide a good overview of the situation facing PHNs in Japan. Is this workforce unionized? If so, what percentage of the workforce belongs to unions? Do workers have the right to create joint management-labor safety and health committees, to address workplace safety and health issues? I am unfamiliar with the labor landscape and law in Japan and these additional details about labor organizing and safety and health committees would be helpful for an international readership. 

Material and Methods

Lines 101 - 103: "In this cross-sectional study, we conducted an anonymous, self-administered questionnaire survey among Japanese PHNs involved in infection prevention and control during the COVID-19 pandemic after January 2020." Please include a copy of questionnaire as an appendix. 

Lines 119-120: "Thus, we extracted items from a qualitative study on distress and its related experience during the pandemic, which we conducted in 2021." Please cite this study, if it is published. 

Results

Tables 3 and 4. These tables are well-organized. However, to improve readability, could you please put the percentages in parentheses? That is, (%) in the headers and (61.3) and (21.5) within the body of the tables? 

It is striking that more than 80% of respondents experienced "Inappropriate staff allocation from outside and within PHCs." Please highlight this finding in these clear terms within the abstract, discussion, and conclusion sections with specific recommendations to improve staffing. 

Discussion

Lines 306 - 311: "During the COVID-19 pandemic, people experienced negative emotions, such as anger, fear, and anxiety [46-48], which were expressed strongly as verbal abuse and complaints toward PHNs. This scenario will definitely be repeated in future health crises. Meanwhile, not all PHNs have the individual capability to accept strong emotions from community residents. Thus, it is an urgent issue to establish an organizational system that can emotionally assist them through sharing their common issues and emotions and offering personal support." This is an important consideration and I agree with your recommendation; however, I wonder if you could make an even stronger recommendation to prevent this maltreatment. Is it possible that management could protect PHNs from maltreatment by placing limitations on patients' abusive behavior? That is, creating consequences for patients and members of the public who mistreat workers, to deter such abuse? I am wondering about the possibility of primary prevention. 

 Lines 317- 320: "However, our survey revealed that more than 80% of PHNs deemed the staff allocation insufficient. Thus, even if organizations developed measures to deal with the situation, the degree of effort can be insufficient for PHNs who are working in the front line." This is a critical finding! Related to this issue, on Lines 349 - 352, you state, "Their burden has to be reduced through system development, collaboration with the community, and staffing personnel distributions. In particular, it is crucial to lessen the burden to prevent burnout in mid-level PHNs, and to stabilize the organization by actively and broadly incorporating their viewpoints and consideration." I agree with this recommendation, given your results, and my knowledge of worker safety and health in the United States. Can you more explicitly highlight this finding and recommendation in the abstract and conclusion sections? That is, express the need for appropriate staffing levels? In the conclusion section, you allude to "appropriate support," but could you explicitly note staffing? In terms of the recommendation to "actively and broadly incorporating their viewpoints and consideration," could you please explain if PHNs belong to unions and can or could participate in worker safety and health committees within their organizations? If you are interested in these issues around burnout within the United States context, I recommend reading this white paper by National Nurses United: https://www.nationalnursesunited.org/campaign/deadly-shame-report

Author Response

15 March 2023

Dear Dr. Yolande Cai

And Reviewers

We sincerely express our gratitude to the Editor and the reviewers who took valuable time and reviewed the manuscript with significant comments. We describe below our point-by-point response to all the comments of the reviewers.

Sincerely,

Akari Miyazaki

Point-by-point response to the reviewers’ comments

Your study is timely and addresses an issue of great practical importance for occupational safety and health, as well as public health more broadly. That is, preventing burnout among Public Health Nurses (PHNs), who are essential to addressing worldwide threats from "disasters, terrorism, and infectious disease." As you note in the Introduction, I wholeheartedly agree that "PHNs being under such pressure, which they cannot carry on their on their own, points to the need to focus on not only individual resilience but also organizational resilience to continue fulfilling their roles in crises."

Your well-written manuscript is excellent, with an appropriate research design, methods that are clearly described, results that are clearly presented, and conclusions that are supported by the results. Prior to publication, I would like to offer a few minor questions, comments, and suggestions for your consideration, in order to further strengthen the manuscript. Below, these comments are organized by manuscript section.

Response:

Thank you very much for warm and considerate comments.

Introduction

You provide a good overview of the situation facing PHNs in Japan. Is this workforce unionized? If so, what percentage of the workforce belongs to unions? Do workers have the right to create joint management-labor safety and health committees, to address workplace safety and health issues? I am unfamiliar with the labor landscape and law in Japan and these additional details about labor organizing and safety and health committees would be helpful for an international readership.

Response:

We really appreciate your valuable advice.

During the current pandemic, there are some movements by Unions of prefectural and municipal workers, Japanese Nursing Association and an individual PHN. As you refer, I agree that those movements are critical to protect themselves, which would indirectly make impact on the community health, as well. Therefore, we added some descriptions about those unions and movements as follows in 1. Introduction.

“According to a study targeting PHNs involved in providing support to people during the recovery stage of the 2011 Fukushima nuclear accident, PHNs played their role with a sense of mission, which is also possible in the current pandemic [27]. However, there were some movements aimed at changing the working environment, which may reflect that the situation was beyond the capability of their strong sense of mission. All-Japan Prefectural and Municipal Worker’s Unions exist in each prefecture and municipality with about 740,000 workers, accounting for approximately 25% of all prefectural and municipal workers. Japanese Nursing Associations are united with 770,000 nurses, about 60% of all working nurses. These unions and associations have been demanding better work environments in PHCs—including securing enough personnel and equipment and higher salaries—based on PHNs’ requests from the national and municipal governments [28,29]. Addressing these demands may be effective in radically reforming the public health system and obtaining support for PHNs. Additionally, in 2021, a PHN reported to the Labour Standards Inspection Office, accusing the municipality of requiring long working hours [30]. Behind all these remarkable movements, there may be a strong desire of PHNs to change the situation to fulfill their own role to protect the community. It is critical to determine the predictors of burnout among PHNs for burnout prevention during a pandemic.

Material and Methods

Lines 101 - 103: "In this cross-sectional study, we conducted an anonymous, self-administered questionnaire survey among Japanese PHNs involved in infection prevention and control during the COVID-19 pandemic after January 2020." Please include a copy of questionnaire as an appendix.

Response:

Thank you for your comment. As you refer, we attached the appendix to show the detail of original items.

Lines 119-120: "Thus, we extracted items from a qualitative study on distress and its related experience during the pandemic, which we conducted in 2021." Please cite this study, if it is published.

Response:

Thank you for your valuable suggestion.

Although we have been hoping that we can cite the article, it is still impossible because it’s still under review in another international journal.

Results

Tables 3 and 4. These tables are well-organized. However, to improve readability, could you please put the percentages in parentheses? That is, (%) in the headers and (61.3) and (21.5) within the body of the tables?

Response:

Thank you for your suggestion. I revised the Table 3 and 4 following your suggestion.

It is striking that more than 80% of respondents experienced "Inappropriate staff allocation from outside and within PHCs." Please highlight this finding in these clear terms within the abstract, discussion, and conclusion sections with specific recommendations to improve staffing.

Response:

Thank you for your valuable comment.

As you mentioned, the point which more than 80% experienced inappropriate staff allocation is important. We added the description to emphasize this point of the result, in abstract, discussion and conclusion;

Abstract

“More than 80% of respondents experienced inappropriate staff allocation.”

“The results highlighted the need for preparing for future health crises, including establishing a system with enough personnel,…”

Discussion (2. Actual State of PHNs’ Experience During the Pandemic)

This shows that the managers who decide policies for PHCs must listen to the PHNs regularly working in the front line to implement adequate measures during each phase of the pandemic.”

Conclusion

“Considering that most PHNs experienced inappropriate staff allocation, it is critical to ensure enough personnel in PHCs located in a community undergoing an outbreak during a future crisis.”

Moreover, to ease the burden of PHNs, personnel arrangement should include securing human resources and sharing roles.”

Discussion

Lines 306 - 311: "During the COVID-19 pandemic, people experienced negative emotions, such as anger, fear, and anxiety [46-48], which were expressed strongly as verbal abuse and complaints toward PHNs. This scenario will definitely be repeated in future health crises. Meanwhile, not all PHNs have the individual capability to accept strong emotions from community residents. Thus, it is an urgent issue to establish an organizational system that can emotionally assist them through sharing their common issues and emotions and offering personal support." This is an important consideration and I agree with your recommendation; however, I wonder if you could make an even stronger recommendation to prevent this maltreatment. Is it possible that management could protect PHNs from maltreatment by placing limitations on patients' abusive behavior? That is, creating consequences for patients and members of the public who mistreat workers, to deter such abuse? I am wondering about the possibility of primary prevention.

Response:

We appreciate your meaningful comment.

In Japan, harassments from customers are social problems in these days. Although the Ministry of Labour and Welfare published the guideline in 2022 to protect employees and workers from harassments including verbal abuse and unreasonable demands and claims from customers, no fundamental resolution (primary prevention) is discussed and suggested in it. No article was found but we made some suggestion referring the resolution of fraud on phone.

“Moreover, as a fundamental resolution, issuing an ordinance to provide a warning and legal move to residents against unreasonable verbal abuse and its dissemination to the wider public may be effective in protecting workers; however, only a few municipalities have such an ordinance in presence [61]. The announcement of “this telephone message is recorded” before the conversation to restrain callers and preventing fraud, which are officially recommended by police departments and used widely, can be implemented in PHCs to prevent unreasonable verbal abuse [62].”

 Lines 317- 320: "However, our survey revealed that more than 80% of PHNs deemed the staff allocation insufficient. Thus, even if organizations developed measures to deal with the situation, the degree of effort can be insufficient for PHNs who are working in the front line." This is a critical finding! Related to this issue, on Lines 349 - 352, you state, "Their burden has to be reduced through system development, collaboration with the community, and staffing personnel distributions. In particular, it is crucial to lessen the burden to prevent burnout in mid-level PHNs, and to stabilize the organization by actively and broadly incorporating their viewpoints and consideration." I agree with this recommendation, given your results, and my knowledge of worker safety and health in the United States. Can you more explicitly highlight this finding and recommendation in the abstract and conclusion sections? That is, express the need for appropriate staffing levels? In the conclusion section, you allude to "appropriate support," but could you explicitly note staffing?

Response:

Thank you for your valuable comment as a great expert in this field.

As you mentioned, the point which more than 80% experienced inappropriate staff allocation is important. We added the description about the need of securing personnel as follows in abstract and conclusion;

Abstract

“The results highlighted the need for preparing for future health crises, including establishing a system with enough personnel,…”

Conclusion

“Considering that most PHNs experienced inappropriate staff allocation, it is critical to ensure enough personnel in PHCs located in a community undergoing an outbreak during a future crisis.”

Moreover, to ease the burden of PHNs, personnel arrangement should include securing human resources and sharing roles.”

In terms of the recommendation to "actively and broadly incorporating their viewpoints and consideration," could you please explain if PHNs belong to unions and can or could participate in worker safety and health committees within their organizations? If you are interested in these issues around burnout within the United States context, I recommend reading this white paper by National Nurses United: https://www.nationalnursesunited.org/campaign/deadly-shame-report

Response:

Thank you for your valuable comment and suggestion. Honestly, before your comment, we did not have such point of view of unions because as you see from the participation rate of unions (25%), it is not very familiar to most Japanese workers of municipalities. However, as your consideration, the unions are playing effective role to protect themselves and it must be important to take a stand on their own issue. So, we added some more concrete suggestion including participation in unions.

“Specifically, participation in unions or associations that can raise the issue with national and municipal governments is one way for an individual worker to protect themselves. Furthermore, as an organization, a PHC should provide opportunities for PHNs to share their problems and distress in a psychologically safe environment [72]. This will enable managers who decide the policies in PHCs to notice others’ ideas and incorporate them in policies, which may result in a more sustainable organization.”

Reviewer 3 Report

The manuscript dealt with PHNs' responses to questionnaires assessing several different components of their experiences with the COVID health crisis. The results were clearly outlined and honestly unsurprising. The pandemic made the job of being a nurse impossible, yet many survived in the job. A few comments:

1. There is a wording error or something (?) wrong on line 55 - "... and triage suspected patients" - not sure what but something doesn't make sense

2. I am bothered by the fact that the authors used their own pre-constructed questionnaires to measure PHNs' experiences and organizational resilience. They did some stats on both, but to a larger degree I have difficulty accepting "in house" questionnaire data.

3. Another issue is the method of data collection, and it's tricky, because I don't have an acceptable answer for how to fix it. The authors know, and actually cite as a limitation, that the sample may be biased, and I agree.

4. Finally, I thought it was noticeable that half of the participating nurses had been working for <10 years - that might be important. I appreciate that the mid-level nurses showed the most negative responding - it might in fact (as the authors state) be because they are teachers and workers, but it might simply be that they're simply more tired of the job.

Author Response

15 March 2023

Dear Dr. Yolande Cai

And Reviewers

We sincerely express our gratitude to the Editor and the reviewers who took valuable time and reviewed the manuscript with significant comments. We describe below our point-by-point response to all the comments of the reviewers.

Sincerely,

Akari Miyazaki

Point-by-point response to the reviewers’ comments

The manuscript dealt with PHNs' responses to questionnaires assessing several different components of their experiences with the COVID health crisis. The results were clearly outlined and honestly unsurprising. The pandemic made the job of being a nurse impossible, yet many survived in the job. A few comments:

1. There is a wording error or something (?) wrong on line 55 - "... and triage suspected patients" - not sure what but something doesn't make sense

Response:

Thank you for your comment. We modified the manuscript.

“In the early stages of the current pandemic, Japanese PHNs faced difficulties in dealing with community residents with negative emotions such as anxiety and anger. It was also distressing to triage suspected patients with limiting testing capacity [14,15].”

2. I am bothered by the fact that the authors used their own pre-constructed questionnaires to measure PHNs' experiences and organizational resilience. They did some stats on both, but to a larger degree I have difficulty accepting "in house" questionnaire data.

Response:

Thank you for your valuable comment. For us, as well, this was the most troubling aspect of our study, so we can understand your concern.

Although we were extremely troubled and repeated discussions, we still came to the conclusion that we just couldn't measure the current situation with the existing measurement scales, but with items which reflect their actual experiences and situations to obtain practical suggestions. Our decision was also supported by the fact that we found that several previous studies have taken these methods (Rosenbäck et al., 2022; Mizota et al.,2006; Omiya and Yamazaki, 2017). We believe that further research on this topic including the appropriateness of this original item is definitely required to obtain more certain and rigid suggestion. To show the significance of this method, we added some description as follows in 2.3.2. PHN experience during the COVID-19 pandemic.

“This method has been utilized by several studies to obtain practical suggestions that reflect the backgrounds of the sample [37-39].”

3. Another issue is the method of data collection, and it's tricky, because I don't have an acceptable answer for how to fix it. The authors know, and actually cite as a limitation, that the sample may be biased, and I agree.

Response:

Thank you for your meaningful comment.

We conducted this research in the midst of the pandemic because it is critical for future pandemic to record and investigate their experience before fading their memories. However, as you referred, at the same time, minimizing the bias is limited in our study. Unless the respondence to our survey was obligated to all PHNs including those who had left their job, we cannot solve this issue of bias. To minimize the bias, we made effort to obtain response from PHNs with various conditions through requesting PHN leader to ensure the awareness to our survey several times.

4. Finally, I thought it was noticeable that half of the participating nurses had been working for <10 years - that might be important. I appreciate that the mid-level nurses showed the most negative responding - it might in fact (as the authors state) be because they are teachers and workers, but it might simply be that they're simply more tired of the job.

Response:

Thank you for your valuable comment.

To address the point that the study subjects tend to have short working experience, it is reflecting the population of PHNs who belong to PHCs in Japan. Japanese PHNs work not only in PHCs but also in other organizational institutions such as central government office and child consultation centers, and PHNs with less experience tends to be assigned in the lower branches (PHCs).

To address the point about the reason of mid-level PHNs with most negative responding, there may be any other facts, as you mentioned. Further research is needed to show what are their distress and motivation toward their job. We added the description in last paragraph of 4.2. Actual State of PHNs’ Experience During the Pandemic as follows;

However, the particularity of mid-level PHNs have not yet been revealed, warranting further investigation.”